# Chromosome territory formation attenuates the translocation potential of cells

Leah F Rosin[†], Olivia Crocker, Randi L Isenhart, Son C Nguyen, Zhuxuan Xu, Eric F Joyce*

Department of Genetics, Penn Epigenetics Institute, Perelman School of Medicine, University of Pennsylvania, Philadelphia, United States

**Abstract** The formation and spatial arrangement of chromosome territories (CTs) in interphase has been posited to influence the outcome and frequency of genomic translocations. This is supported by correlations between the frequency of inter-chromosomal contacts and translocation events in myriad systems. However, it remains unclear if CT formation itself influences the translocation potential of cells. We address this question in *Drosophila* cells by modulating the level of Condensin II, which regulates CT organization. Using whole-chromosome Oligopaints to identify genomic rearrangements, we find that increased contact frequencies between chromosomes due to Condensin II knockdown leads to an increased propensity to form translocations following DNA damage. Moreover, Condensin II over-expression is sufficient to drive spatial separation of CTs and attenuate the translocation potential of cells. Together, these results provide the first causal evidence that proper CT formation can protect the genome from potentially deleterious translocations in the presence of DNA damage.
DOI: https://doi.org/10.7554/eLife.49553.001

*For correspondence:
erjoyce@upenn.edu

Present address: [†]Nuclear Organization and Gene Expression Section, Laboratory of Cellular and Developmental Biology, National Institute of Diabetes and Digestive and Kidney Diseases, National Institutes of Health, Bethesda, United States

Competing interests: The authors declare that no competing interests exist.

## Introduction

Chromosomes undergo an elaborate folding pattern in the interphase nucleus, ultimately occupying distinct domains known as chromosome territories (CTs) (*Bickmore and van Steensel, 2013*; *Ciabrelli and Cavalli, 2015*; *Sexton and Cavalli, 2015*). The partitioning of chromosomes into CTs limits inter-chromosomal interactions in the nucleus. CT formation has been observed across a wide range of species and cell types by both fluorescence in situ hybridization (FISH) and chromatin conformation capture-based techniques (*Bauer et al., 2012*; *Branco and Pombo, 2006*; *Cremer and Cremer, 2001*; *Cremer and Cremer, 2010*; *Pecinka et al., 2004*; *Rosin et al., 2018*; *Smeets et al., 2014*; *Tanabe et al., 2002*). However, despite the widespread prevalence and conservation of CTs, the molecular determinants and function of this level of organization has remained elusive.

Notably, levels of inter-chromosomal contact between different chromosome pairs have been correlated with increased translocation frequencies – both those occurring naturally in the human population and those induced experimentally in mammalian cells (*Arsuaga et al., 2004*; *Bickmore and Teague, 2002*; *Branco and Pombo, 2006*; *Canela et al., 2017*; *Engreitz et al., 2012*; *Hlatky et al., 2002*; *Holley et al., 2002*; *Klein et al., 2011*; *Roukos et al., 2013*; *Zhang et al., 2012*). These data support a 'contact-first' model of translocation genesis (*Aten et al., 2004*; *Engreitz et al., 2012*; *Foster et al., 2013*; *Meaburn et al., 2007*; *Savage, 1998*; *Savage, 2000*), arguing that CT formation and positioning in the nucleus can influence the outcome and frequency of chromosomal translocations (*Roukos et al., 2013*; *Soutoglou and Misteli, 2008*). However, while these correlative data support a role for inter-chromosomal contacts in directing the location of translocations, it remains unclear if translocations are attenuated by CT formation itself.

*Drosophila* cells provide a unique opportunity to directly test the role of CT partitioning in translocation genesis. Similar to human chromosomes, the three major *Drosophila* chromosomes form robust CTs throughout interphase that can be labeled simultaneously with Oligopaint-based chromosome paints (*Nguyen and Joyce, 2019*; *Rosin et al., 2018*). Additionally, genome-wide chromosome paints provides a robust system to perform karyotype analysis in parallel and quantify the absolute frequency of translocation events in a cell population. Finally and most importantly, the extent to which chromosomes are packaged into CTs can be modulated in *Drosophila* cells by altering the activity of Condensin II, a highly conserved SMC protein complex that is essential for large-scale chromosome folding and proper CT formation (*Bauer et al., 2012*; *Li et al., 2015*; *Rosin et al., 2018*). Here, we use this system to explore the causal relationship between CT partitioning and translocation frequency.

## Results

### Whole-chromosome oligopaints can efficiently detect IR-induced translocations

Our previous work demonstrated that Oligopaint labeling of whole chromosomes during interphase is sufficiently sensitive to detect stable translocation events in the cell population (*Rosin et al., 2018*). We observed that preferential CT positioning in different *Drosophila* cell lines corresponds to stable translocations found in those cell populations (*Rosin et al., 2018*). To determine if we could detect induced translocations that are more rare and varied in size, we turned to *Drosophila* BG3 cells which are derived from the central nervous system of third-instar larvae and maintain a diploid karyotype with infrequent spontaneous rearrangements (*Rosin et al., 2018*). To create DNA double-strand breaks (DSBs) and induce translocations, we subjected BG3 cells to either a low dose (5 Gy) or high dose (20 Gy) of ionizing irradiation (IR). We found that most cells recovered by 48 hr after IR in both conditions based on a reduction in γ-H2Av staining, which marks sites of DSBs (*Figure 1—figure supplement 1*) (*Mehrotra and McKim, 2006*). Neither 5 Gy nor 20 Gy treatments significantly altered cell viability or cell population growth (*Figure 1—figure supplement 1*).

To identify translocations, cells were arrested in metaphase 48 hr after IR and karyotyped using our whole-chromosome Oligopaints labeling chromosomes X, 2, and 3 (*Figure 1A*). This strategy allowed us to quantify the color junctions that form as the result of translocation events and measure their frequency between each chromosome pair. Because this analysis is performed on a single-cell basis, these translocation junctions can be easily identified regardless of whether recurrent or variable breakpoints occur throughout the cell population. A total of 1402 metaphase spreads were scored for translocations across 3–5 biological replicates. In each replicate, we found that translocations were efficiently produced and detected following exposure to both 5 Gy and 20 Gy IR, with 3% and 14.8% of total cells harboring a translocation, respectively (*Figure 1B*). We also found a few cases of spontaneous translocations in untreated cells (1.7%). Translocations between all chromosome pairs were recovered after 20 Gy IR, which we sub-classified as discrete translocations (mid-arm translocations where only two chromosomes were involved; 60.2%), compound chromosomes (fusions of seemingly whole chromosome arms from two different chromosomes; 32%), and complex rearrangements (resulting from multiple translocation events; 7.8%; *Figure 1A–B*). Approximately 33% of translocations were reciprocal with a seemingly equal exchange of genetic material between the two chromosomes involved (*Figure 1—figure supplement 1*).

Notably, different chromosome pairs exhibited different frequencies of translocations, with most translocations occurring between chromosomes 2 and 3 (*Figure 1C*). As expected, the different frequencies of translocations between chromosomes can largely be explained by the total genomic length of the chromosomes involved (p=0.0005, $r^2 = 0.72$, *Figure 1D*). However, there were variations in translocation frequency across the biological replicates for each chromosome pair. In particular, translocations between chromosomes 2 and 3 varied in frequency from 7% to 14% of the cell population depending on the replicate (*Figure 1C–D*). This suggests that factors other than chromosome size might be contributing to their translocation potential.

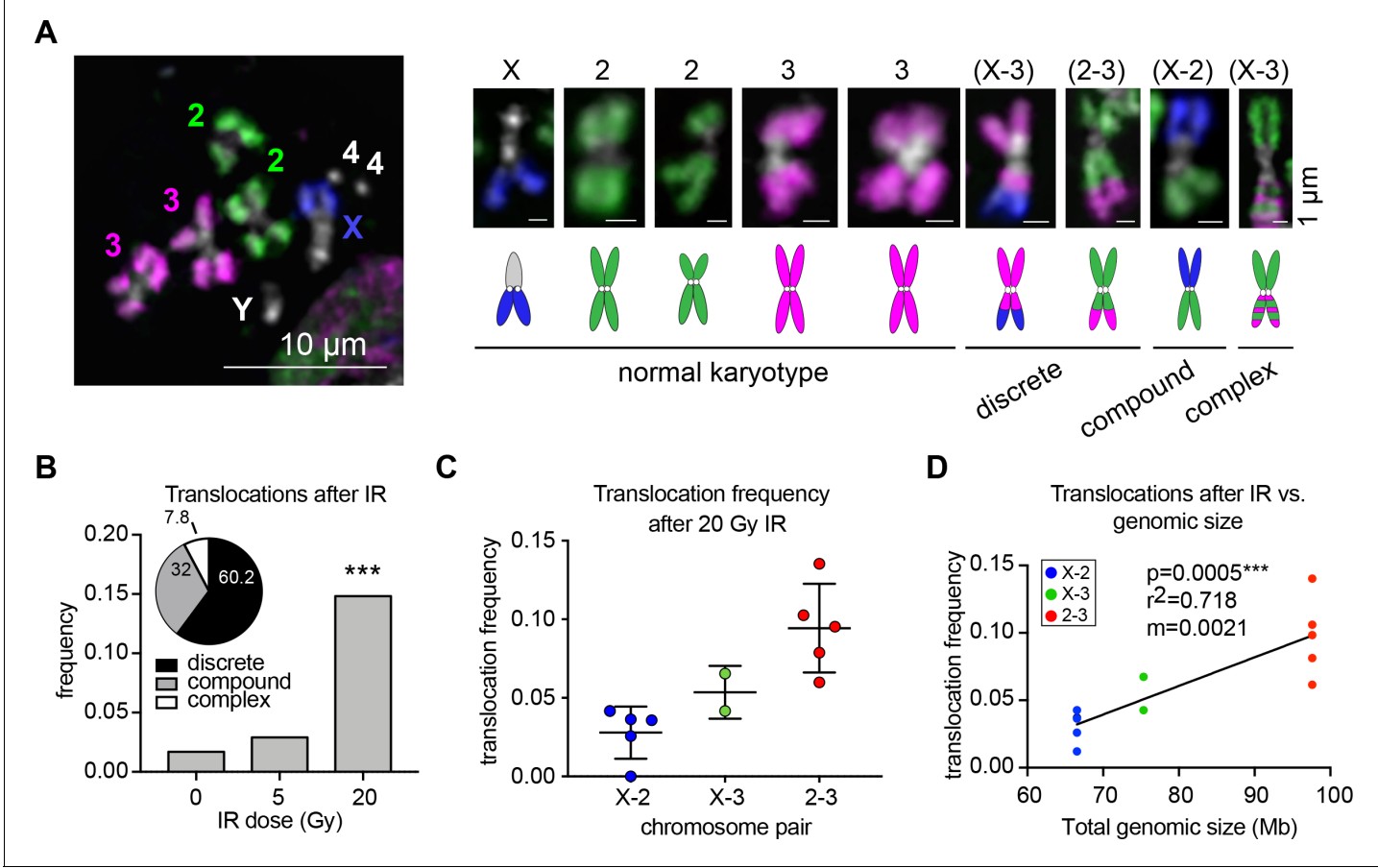

**Figure 1.** Whole-chromosome Oligopaints can efficiently detect IR-induced translocations. (A) Left: representative metaphase spread with chromosome paints in control BG3 cells. DNA is stained with Hoechst and is shown in white. Right: representative chromosomes 48 hr after irradiation. Both normal and rearranged chromosomes are shown, with cartoon schematics of the chromosomes directly below. The chromosomes involved in the rearrangement (if any) are listed above, and the classification of each translocation type is listed below. (B) Total translocation frequency after varying doses of IR for 3–5 biological replicates. n = 592, 368, and 442 spreads counted for no IR, 5 Gy, and 20 Gy, respectively. Inset: Pie graph depicting the relative translocation types identified after 20 Gy of IR as a percent of total translocations. (C) Dot plot showing translocation frequencies for 2–5 populations of cells 48 hr after 20 Gy IR treatment. Only two replicates were included for X-3 due to pre-existing translocations in those cell sub-populations. (D) Scatterplot showing the translocation frequency after 20Gy IR (Y-axis) versus total genomic size of the chromosome pair (X-axis). The data shown represent five biological replicates. m = slope of line of best fit. P-value was calculated by linear regression.

DOI: https://doi.org/10.7554/eLife.49553.002

The following figure supplement is available for figure 1:

**Figure supplement 1.** Additional data related to *Figure 1*.

DOI: https://doi.org/10.7554/eLife.49553.003

## Inter-CT contact frequency correlate with translocation frequency in *Drosophila*

A positive correlation between the extent of CT intermixing and translocation frequency by FISH has been observed in human lymphocytes (*Branco and Pombo, 2006*). Therefore, we sought to determine if the variation in translocation frequencies between chromosome pairs and replicates is influenced by varied intermixing between neighboring CTs in different populations of BG3 cells. We created an experimental scheme in which a single population of cells was divided into three groups. One group was immediately fixed and subjected to interphase FISH to analyze CT positioning prior to IR treatment. The other groups were either subjected to IR or left untreated as a no IR control and subsequently karyotyped by whole-chromosome painting to quantify translocations (*Figure 2A*). This experiment was repeated five independent times to thoroughly capture the range of CT

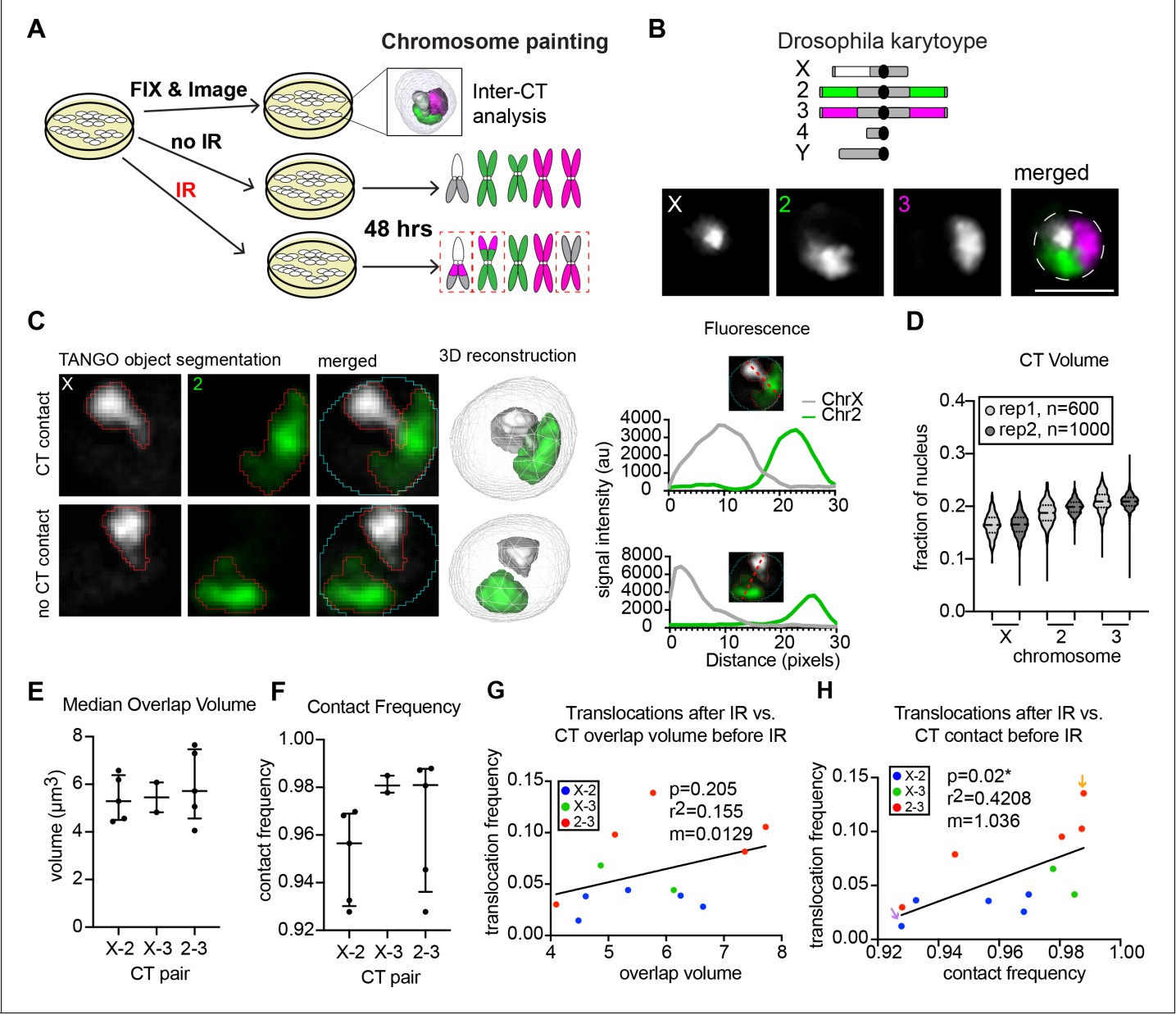

**Figure 2.** Inter-CT contacts correlate with translocation frequency in *Drosophila*. (A) Schematic of experimental design in which cells are split into three groups: one of which is harvested immediately for interphase FISH analysis, the second is subjected to IR treatment, and the third is used as a 'no IR' control. The latter two groups are allowed to recover for two additional days before karyotype analysis to identify translocations (dashed red boxes). (B) Top: Cartoon depiction of BG3 cell karyotype and chromosome paints. Unlabeled heterochromatin is shown in gray. Bottom: representative nucleus with Oligopaints labeling chromosome X (white), 2 (green), and 3 (magenta). Dotted line in merged image represents the nuclear edge. Scale bar equals 5 μm. (C) Left: Representative image showing chromosome X paint (white) and chromosome two paint (green) in two representative nuclei illustrating CT contact and no CT contact. CT segmentation is shown as a red outline. Final 3D rendering is shown on the right. Right: Line plots of fluorescence intensity from the two cells depicted on left. In the top graph, voxel colocalization is observed while there is no voxel colocalization in the bottom graph. (D) Violin plot of CT volume for chromosomes X, 2, and three as a fraction of nuclear volume in BG3 cells. Each violin represents a single cell population, and two biological replicates are shown, each with >500 nuclei being measured. (E) Dot plot showing the median CT overlap volume between chromosome pairs defined by the X-axis, for 2–5 cell populations, where each dot represents the median of a cell population of n > 500 cells. (F) Dot plot showing the fraction of cells with CT contact between chromosome pairs defined by the X-axis, for 2–5 cell populations, where each dot represents the average of a cell population of n > 500 cells. (G) Scatterplot showing the translocation frequency after 20Gy IR (Y-axis) versus median CT overlap volumes prior to IR (X-axis). The data shown represent 3–5 biological replicates. m = slope of line of best fit. P-value was calculated by linear regression. (H) Scatterplot showing the translocation frequency after 20Gy IR (Y-axis) versus inter-CT contact frequency prior to IR (X-axis). The data

*Figure 2 continued on next page*

*Figure 2 continued*

shown represent 3–5 biological replicates. m = slope of line of best fit. P-value was calculated by linear regression. Arrows depict populations with the lowest (purple) and highest (yellow) translocation events.

DOI: https://doi.org/10.7554/eLife.49553.004

The following figure supplement is available for figure 2:

**Figure supplement 1.** Additional data related to *Figure 2*.

DOI: https://doi.org/10.7554/eLife.49553.005

positioning during interphase prior to IR and the translocation frequencies across different cell populations.

For interphase CT organization, Oligopaints labeling chromosomes X, 2, and three were segmented by custom image analysis to trace the 3D edges of each chromosomal FISH signal (*Figure 2B,C*) (*Ollion et al., 2013*; *Rosin et al., 2018*). We observed similar average volumes for all chromosomes, with chromosome X being slightly smaller than chromosomes 2 and 3 (18.9–20.4 $\mu m^3$ for chromosome X, 22.8–23.1 $\mu m^3$ for chromosome 2, and 24.1–25.9 $\mu m^3$ for chromosome 3), and minimal variability in the range of CT volumes between biological replicates (*Figure 2D*). Notably, we did not find any cells where a single chromosome occupied more than 28% of the nucleus, indicating that CTs are stably compacted in BG3 cells. (*Figure 2D*).

We next measured the extent of CT intermixing and inter-CT contact frequencies for all chromosome pairs. CT intermixing was defined as the volume of voxel colocalization between two neighboring chromosomes and is measured on a cell-by-cell basis. Inter-CT contact frequency is the percentage of cells in which the chromosome pair exhibited at least a single voxel of colocalization (*Figure 2C*). The median intermixing volume between all chromosome pairs was minimal, ranging from 4 to 8 $um^3$, which represents ~5% of the nucleus (*Figure 2E*). This is consistent with previous Hi-C studies of CTs showing that *trans* interactions make up only ~7% of total chromosome contacts in *Drosophila* (*Li et al., 2015*). However, despite minimal CT intermixing volumes, our data show that CTs are in close proximity and contact each other in 93–99% of BG3 cells, depending on the chromosome pair and replicate line being examined (*Figure 2F*). This is consistent with our data from tetraploid Kc167 cells (*Rosin et al., 2018*), showing that *Drosophila* chromosomes exhibit high levels of CT contact independent of their ploidy and karyotype. Following exposure to 5 Gy or 20 Gy IR, there were no significant changes in nuclear volume, CT volume, or inter-CT contact frequencies in BG3 cells (*Figure 2—figure supplement 1*).

To determine if the extent of intermixing between different CTs influences translocation potential, the median CT intermixing volume between chromosome pairs prior to IR in each replicate line was then plotted against the frequency of their translocations measured after exposure to 20 Gy. Surprisingly, we found only a weak correlation between intermixing volume and translocation frequency for each chromosome pair and replicate population ($r^2$ = 0.16, p=0.21; *Figure 2G*). Instead, we found a significant positive correlation between the frequency of inter-CT contacts and translocation frequency ($r^2$ = 0.42, p=0.02), such that higher frequencies of inter-CT contact increase the chances of translocation events occurring between those two chromosomes (*Figure 2H*). For example, the chromosome pair and population with the lowest contact frequency overall (X-2, 92.5% contact) had the lowest number of translocations after IR (*Figure 2H*, purple arrow), whereas the chromosome pair and population with the highest CT contact frequency overall (2–3, 99% contact) harbored a translocation in nearly 14% of cells after IR (*Figure 2H*, orange arrow). Indeed, a linear regression analysis of the data predicts a 1% increase in translocation frequency for every 1% increase in contact frequency. This indicates that efficient translocation formation requires a high frequency of CT contact and yet is extremely sensitive to subtle differences in inter-CT contact frequencies across cell populations. Considering that increased intermixing between two chromosomes does not predict an increase in translocation frequency, additional points of contact between chromosomes do not necessarily increase the likelihood of translocation formation.

## CT disruption following knockdown of Cap-H2 increases the translocation potential of long chromosomes

Our findings in BG3 cells suggest that higher contact frequencies between two chromosomes in the nucleus increase translocation potential. To further test this model, we next sought to abrogate the activity of the Condensin II complex in *Drosophila*, which has been shown to increase inter-chromosomal interactions as observed by Hi-C and FISH (*Figure 3A–C*) (*Bauer et al., 2012*; *Li et al., 2015*; *Rosin et al., 2018*). Using publically available Hi-C data from Kc167 cells (*Li et al., 2015*) depletion of the Condensin II subunit Cap-H2 leads to a 20–27% decrease in cis interactions and a 7–17% increase in trans interactions depending on the chromosome pair (*Figure 3A–B*). Consistent with this, Oligopainting of chromosomes X, 2, and 3 in BG3 cells following Cap-H2 knockdown revealed significant increases in chromosome volume, CT intermixing, and inter-CT contact frequencies between all chromosome pairs (*Figure 3C–F* and *Figure 3—figure supplement 1*). Note, however, that the frequency of chromosome contacts by FISH is already high in untreated BG3 cells and was therefore only increased by 2–5% following Cap-H2 knockdown (*Figure 3E*). No corresponding defects in chromosome segregation or viability were detected (*Figure 3—figure supplement 1*), consistent with previous reports suggesting Condensin II is dispensable for mitosis in *Drosophila* cells (*Hartl et al., 2008*; *Rosin et al., 2018*; *Savvidou et al., 2005*). FISH-based karyotype analysis also revealed no significant increase in translocation frequency or change in ploidy after 4 days of Cap-H2 knockdown (*Figure 3—figure supplement 1*) (*Rosin et al., 2018*). Furthermore, DNA damage was not increased after Cap-H2 knockdown alone compared to controls based on γ-H2AV immunostaining (*Figure 3—figure supplement 1*). Therefore, CT decompaction and increased intermixing due to Cap-H2 depletion alone are likely not causal events in acute genome instability.

To determine if increased inter-chromosomal interactions as a result of Cap-H2 depletion increases translocation potential in the presence of DNA damage, we repeated our IR experimental scheme in triplicate following RNAi depletion of Cap-H2. Following 5 Gy and 20 Gy of IR, we noted that DNA repair kinetics were delayed following depletion of Cap-H2; however, similar to control cells, most DSBs were repaired by 48 hr post-IR (*Figure 3—figure supplement 1*). No significant changes in cell viability, ploidy, or the frequency of chromosome fragments were observed after 20 Gy IR (*Figure 3—figure supplement 1*) indicating that Cap-H2 depleted cells continued to cycle following IR and chromosome breaks were undergoing repair.

The overall frequency of cells with chromosomal translocations increased, albeit moderately, 48 hr post-IR with either 5 Gy or 20 Gy treatments (*Figure 3G*). When analyzing chromosome pairs separately, chromosomes 2 and 3 exhibited a significant 50% increase in their translocation frequency following 20 Gy IR across all three replicates (*Figure 3H*). We considered the possibility that we were missing some rearrangements from DSBs that were repaired between homologous chromosomes due to increased homolog pairing following Cap-H2 depletion. However, the translocation frequency of the X chromosome, which lacks a homologous partner in male diploid BG3 cells, again only showed a moderate increase in translocation frequencies following IR treatment and Cap-H2 depletion (*Figure 3H*). This suggests that DSB repair between homologous, versus heterologous, chromosomes is not increased following Cap-H2 depletion. Additionally, in the small subset of tetraploid cells with two X chromosomes, the presence of a pairing partner for the X chromosome does not reduce the frequency of X-2 translocations in either control or Cap-H2-depleted cells (*Figure 3—figure supplement 1*). In contrast, X-2 translocation frequency is significantly increased when two X chromosomes are present, consistent with our observation above that translocation frequency trends with total genomic content.

We conclude that Cap-H2 depletion can increase the likelihood of forming heterologous translocations in the presence of DSBs, particularly between the two largest chromosomes. The moderate increase in translocation events mimics the small increase in chromosome contact frequencies between control and Cap-H2-depleted cells, further supporting the idea that inter-CT contact frequencies are predictive of translocation potential.

## Increased condensin II activity can attenuate the potential to form translocations in the presence of DNA damage

In *Drosophila*, increasing levels of Condensin II by direct over-expression of the limiting Cap-H2 subunit results in smaller CTs that are more spatially separated from each other during interphase

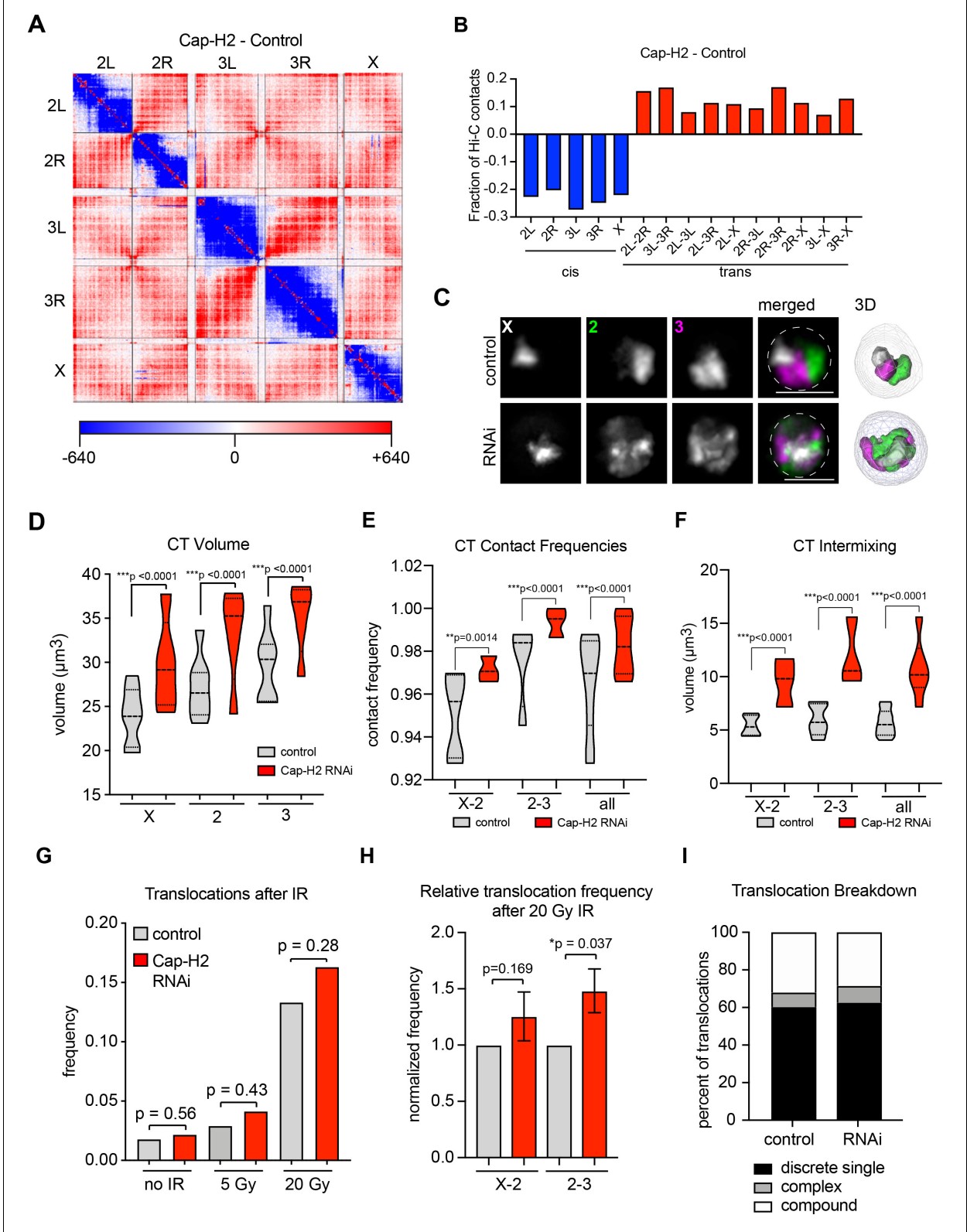

**Figure 3.** CT disruption following knockdown of Cap-H2 increases the translocation potential of long chromosomes. (**A**) Whole genome heat map obtained by subtracting the two-dimensional contact matrix of Hi-C data from control and Cap-H2 depleted Kc167 cells. Hi-C data obtained from *Li et al. (2015)*. (**B**) Bar graph showing the intra-chromosomal and inter-chromosomal changes for each chromosome pair calculated from the Cap-H2-Control Hi-C subtraction map. (**C**) Representative nucleus with Oligopaints labeling chromosome X (white), 2 (green), and 3 (magenta) in control

*Figure 3 continued on next page*

Figure 3 continued
conditions (top), or after Cap-H2 RNAi (RNAi; bottom). Dotted line in merged image represents the nuclear edge. Scale bar = 5 μm. Right: 3D rendering of segmented chromosome structures. (D) Violin plot showing average CT volumes across three biological replicates, both before and after IR, where n > 500 cells each, for control and Cap-H2 RNAi. p-values were determined by Student's t-test. (E) Violin plot showing CT contact frequencies for X-2 and 2–3 CT pairs and combined across three biological replicates for control and Cap-H2 RNAi. p-values were determined by a Fisher's Exact Test comparing contact and no contact for individual replicates. (F) Violin plot showing CT intermixing volumes for X-2 and 2–3 CT pairs and combined across three biological replicates for control and Cap-H2 RNAi. p-values were determined by Student's t-test. (G) Bar graph showing the total translocation frequency after no IR, 5 Gy IR, 20 Gy IR, and combined ('all') for control and Cap-H2 RNAi cells. P-values were calculated by Fisher's exact test comparing normal karyotypes to those with translocations for control and RNAi. (H) Bar graph showing fold-change in translocation frequencies of control and Cap-H2 RNAi cells after 20Gy IR. All data are normalized to controls, with controls shown in gray and Cap-H2 RNAi in red. p-values were calculated using Fisher's exact test. (I) Stacked bar graphs showing the types and frequency of translocations in control and Cap-H2 RNAi cells after 20 Gy IR. n = 47 (control) and 56 (Cap-H2 RNAi) cells with translocations.
DOI: https://doi.org/10.7554/eLife.49553.006
The following figure supplement is available for figure 3:

**Figure supplement 1.** Additional data related to *Figure 3*.
DOI: https://doi.org/10.7554/eLife.49553.007

(*Buster et al., 2013*; *Rosin et al., 2018*). To determine if reduced inter-CT contact in BG3 cells would lead to a corresponding decrease in translocation potential, we generated a stable cell line that can be rapidly induced to overexpress Cap-H2 (OX) (*Figure 4A*). Following induction, Oligopaint FISH targeting chromosomes X, 2, and 3 confirmed the formation of CTs that are more compact and spatially separated from each other compared to uninduced controls (*Figure 4B*). Note that homologs were also more frequently unpaired, resulting in two CTs per chromosome (*Figure 4B*). Nevertheless, the total CT volume per nucleus was reduced compared to controls (*Figure 4C*). A significant overall reduction in inter-CT contact frequencies and CT intermixing volume was also observed (*Figure 4D–E*).

Following overexpression, cells were harvested for interphase FISH or exposed to 20 Gy of IR and karyotyped after a 48 hr recovery period, as described above. Remarkably, the percentage of cells harboring a chromosomal translocation was significantly reduced from 16.7% in uninduced cells to 10.9% after Cap-H2 OX induction (p=0.02), representing a 35% reduction in translocation frequency (*Figure 4F*). This reduction was more dramatic when examining data from chromosome pairs separately. In particular, X-2 and 2–3 translocation frequencies were reduced by ~50% each (*Figure 4G*). There was no significant change in the distribution of rearrangement types (discrete, compound and complex), suggesting that all types of translocations were equally reduced (*Figure 4H*). Finally, inter-CT contact frequencies between specific chromosome pairs and populations remained significantly correlated with translocation frequency following Cap-H2 overexpression ($r^2$ = 0.6033, p=0.014; *Figure 4I*). Indeed, a nearly 1:1 change in the percentage of chromosome contact and translocation frequency was maintained.

Importantly, no significant decrease in DNA damage was seen in Cap-H2 OX cells following IR compared to controls, suggesting that the altered chromatin morphology as a result of excess Condensin II does not itself protect the cell from damage (*Figure 4J*). Also, no defects in DNA repair kinetics or viability were observed in Cap-H2 OX cells compared to controls (*Figure 4J–K*), indicating that Cap-H2 OX does not acutely impact DNA repair or cell survival within these time-points. There was also no observable increase in chromosome segregation defects during anaphase in Cap-H2 OX cells compared to controls, either before or after IR (*Figure 4—figure supplement 1*). Finally, no significant change was seen in cell ploidy, the amount of reciprocal translocations, or the quantity of dicentric or acentric chromosomes in Cap-H2 OX cells compared to controls (*Figure 4—figure supplement 1*). Together, these results suggest that mitosis and cell cycle progression are unaffected by Cap-H2 OX.

For independent confirmation of the above results, we genetically induced upregulation of Condensin II activity by depleting BG3 cells of the SLMB ubiquitin ligase component that directly targets Cap-H2 for degradation (*Figure 4—figure supplement 1*) (*Buster et al., 2013*). Following IR treatment, SLMB-depleted cells also showed a significant reduction in CT contact frequencies and a > 50% reduction in translocation formation compared to controls (*Figure 4—figure supplement 1*). No significant defects in DNA repair kinetics or viability were observed following IR in SLMB

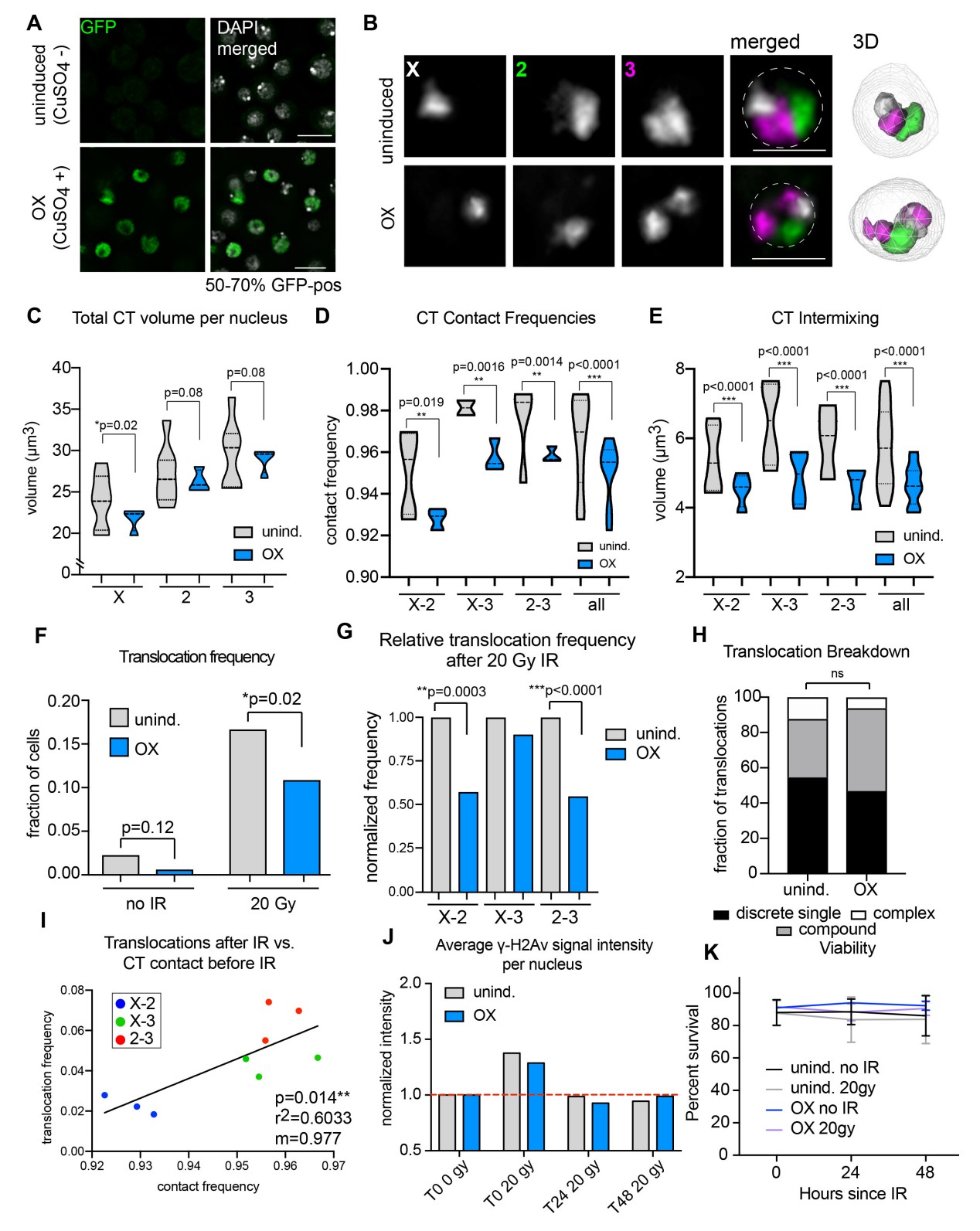

**Figure 4.** Increased Condensin II activity can attenuate the potential to form translocations in the presence of DNA damage. (A) Immunofluorescence showing Cap-H2-GFP expression levels after induction (or uninduced, top row). DAPI is shown in gray. GFP is shown in green. Scale bar = 10 μm. (B) Left: representative nuclei with Oligopaints labeling chromosome X (white), 2 (green), and 3 (magenta) in control conditions (top), or after Cap-H2 overexpression (OX; bottom). Dotted line in merged image represents the nuclear edge. Scale bar = 5 μm. Right: 3D rendering of segmented

*Figure 4 continued on next page*

*Figure 4 continued*

chromosome structures. (C) Violin plot showing average CT volumes across three biological replicates, where n > 500 cells each, for control and Cap-H2 OX. P-values were calculated by Student's t-test. (D) Violin plot showing CT contact frequencies for all CT pairs and combined across three biological replicates for control and Cap-H2 OX. p-values were determined by a Fisher's Exact Test comparing contact and no contact for individual replicates. (E) Violin plot showing CT intermixing volumes for all CT pairs grouped together across three biological replicates for control and Cap-H2 OX. p-values were determined by Student's t-test. (F) Total translocation frequency before or after 20Gy IR for control and Cap-H2 OX cells. p-values were calculated by Fisher's exact test comparing normal karyotypes to those with translocations for control and OX. (G) Fold-change in translocation frequencies of control and Cap-H2 OX cells after 20Gy of IR. All data are normalized to controls, with uninduced controls being shown in gray and Cap-H2 OX shown in blue. P-values were calculated using Fisher's exact test. (H) Stacked bar graphs showing the types and frequency of translocations in control and Cap-H2 OX cells after 20 Gy IR. n = 54 (control) and 63 (Cap-H2 OX) cells with translocations. p>0.5; calculated by Fisher's exact test comparing control to OX for each category. (I) Scatterplot showing the translocation frequency of Cap-H2 OX cells after 20 Gy IR (Y-axis) versus CT contact frequencies before IR (X-axis). The data shown represent three biological replicates. m = slope of line of best fit. $r^2$ and p values were calculated by linear regression. (J) Quantification of anti-γ-H2Av staining on control (gray) or Cap-H2 OX cells (blue) before IR (T0 0 Gy), immediately after 20 Gy IR (T0 20 Gy), 24 hr (T24 20 Gy), and 48 hr (T48 20 Gy). (K) Line graph showing average cell viability in control or Cap-H2 OX cells before and after 20 Gy IR, measured by trypan blue staining. Error bars show standard deviation between biological and technical replicates.

DOI: https://doi.org/10.7554/eLife.49553.008

The following figure supplement is available for figure 4:

**Figure supplement 1.** Additional data related to *Figure 4*.

DOI: https://doi.org/10.7554/eLife.49553.009

knowdown cells compared to controls (*Figure 4—figure supplement 1*). Taken together, these results illustrate that the spatial separation of CTs driven by Condensin II activity can attenuate the potential to form translocations in the presence of DNA damage.

## Discussion

In summary, we show that chromosomes can be visualized and karyotyped for rare and varied translocations using chromosome-wide Oligo-based chromosome paints (*Nguyen and Joyce, 2019*). In contrast to population-based methods that can measure relative translocation breakpoint usage, chromosome painting offers absolute translocation frequency in the cell population. We find that translocation frequencies in *Drosophila* cells strongly correlate with chromosome size, and variations in translocation frequencies between cell populations can be explained by changes in inter-CT contact frequency. Indeed, in the presence of DNA damage, a 1% increase in inter-CT contact in a cell population yields a 1% increase in translocation frequency. Surprisingly, the extent of intermixing between chromosomes does not correlate with translocation potential, in contrast to what has been observed in human cells (*Branco and Pombo, 2006*). This discrepancy could be explained by the relatively larger number of chromosomes in human cells, which would limit the probability of contact between any two chromosomes in the nucleus. In this context, the median intermixing volume measured between each chromosome pair may be completely dependent on the frequency of their contact in the cell population, with an increased number of cells in contact yielding a higher median intermixing volume. This would support a model in which inter-CT contact frequencies in the cell population are the predominant spatial risk factor for translocation genesis, consistent with previously observed correlations between preferential CT neighbors and translocation frequency in a number of systems (*Arsuaga et al., 2004*; *Branco and Pombo, 2006*; *Engreitz et al., 2012*; *Hlatky et al., 2002*; *Holley et al., 2002*; *Parada et al., 2004*; *Zhang et al., 2012*).

What has remained unclear is whether a loss of CT integrity would lead to increased translocation potential. Using Condensin II depletion as a genetic tool to disrupt CT partitioning in *Drosophila* cells, we establish that increasing inter-chromosomal interactions does indeed predispose large chromosomes to increased translocations in the presence of DNA damage. Moreover, by overexpressing Condensin II directly or by removing its negative regulator SLMB, we are able to enhance the spatial separation of CTs prior to IR and demonstrate that this is sufficient to reduce the translocation potential of chromosome pairs by up to 50%. Importantly, we cannot rule out the possibility that Condensin II impacts translocation frequency independent from its role in CT formation. For example, genome instability due to altered chromatin structure or chromosome segregation defects could contribute to translocation genesis. However, we do not observe any increased DNA damage or chromosome segregation defects in *Drosophila* BG3 cells following either Condensin II depletion

or overexpression. Moreover, we observed a 1:1 decrease in inter-CT contact and translocation frequency between specific chromosome pairs following Cap-H2 overexpression. Therefore, these data are more consistent with a causal role of CT organization in maintaining genome integrity. Considering the only method currently available to modulate CT organization is through altering Condensin II activity, further testing of this model awaits the identification of additional factors.

Although Condensin II is essential for proper CT compaction in *Drosophila* and yeast (*Iwasaki et al., 2016*; *Rosin et al., 2018*), it remains unclear if this function is conserved in mammals due to the essential role of this complex in chromosome segregation. Intriguingly, however, loss of the SWI/SNF chromatin remodeling complex component ARID1A in human cells has recently been shown to disrupt NCAPH2 localization during interphase and lead to increased chromosome volumes and inter-CT interactions by both Hi-C and FISH (*Wu et al., 2019*). Moreover, NCAPH2 loss significantly negatively correlated with increased inter-chromosomal interactions observed in the Hi-C analysis, suggesting Condensin II's role in CT partitioning during interphase may be conserved in human cells (*Wu et al., 2019*).

It is also worth noting that Condensin II has been linked to a number of diseases, including cancer (*Ham et al., 2007*; *Leiserson et al., 2015*; *Wang et al., 2018*). In particular, both loss of Condensin II subunits and disruption of their loading onto chromatin in mice has been reported to drive lymphomagenesis with highly rearranged chromosomes in the transformed cells (*Atchison, 2014*; *Ishak et al., 2017*; *Martin et al., 2016*; *Woodward et al., 2016*). While the translocations in these cases may result from chromosome segregation defects, it remains possible that the spatial separation of interphase chromosomes is a novel pathway by which the Condensin II complex promotes genome stability. In this context, *Drosophila* Condensin II offers a unique opportunity to directly study the role of this complex in chromosome folding independent of its role in chromosome segregation, allowing us to investigate how this specific function of Condensin II impacts genome integrity. In the future, it will be interesting to examine this system in vivo in combination with recently developed sequencing-based platforms (*Chiarle et al., 2011*; *Klein et al., 2011*) to determine if particular tissues or genomic sites are sensitive to either Condensin II depletion or overexpression.

## Materials and methods

### Cell lines and tissue culture

BG3 (DGRC 166) cells were obtained from the *Drosophila* Genome Resource Center and were grown at 25°C in M3 media, supplemented with 10% FBS and 10 μg/ml insulin. To ensure that experiments were done with log-phase cells, active cultures were split at a 1:4 ratio twice per week, and passaged at $2 \times 10^6$ cells/mL 24 hr prior to experiments. For Cap-H2 overexpression experiments, a pMT-Cap-H2::GFP construct was stably integrated into BG3 cells by co-transfecting with a hygromycin selection plasmid. Cap-H2 was induced with 0.5 mM CuSO$_4$ for 24 hr.

### RNAi-mediated knock down in cultured cells

The following primers were used for T7 RNA synthesis:

| dsRNA target | F primer | R primer |
| --- | --- | --- |
| Brown (control) | CTATGGCGTGACGTATATATTT | GATATTATCGATGTCGATCCAG |
| Cap-H2 | GAGCACATGACCACAAAGG | TATGCATTTGAATATCGGAAAG |
| slmb | CACCAGGCGATCTCTGTA | ACACTGGATCGGTGCTGT |

dsRNA was generated using the MegaSCRIPT T7 kit (Applied Biosystems) and purified using the RNeasy Kit (Qiagen). Application of RNAi to cells was carried out by soaking in serum-free media according to published methods (*Ramadan et al., 2007*). Briefly, for RNAi in a 6-well plate, $2 \times 10^6$ cells were incubated with 20 μg of dsRNA in 1 mL of serum-free medium for 30 min. After incubation, 2 mL of serum-containing medium was added to cells, followed by incubation for 4 days, or retreated every 3–4 days for the Cap-H2 extended RNAi.

## Irradiation

After 18–24 hr of copper sulfate treatment (or water for uninduced controls), cells were irradiated with either 5 or 20 gamma rays, using a Cs-137 Gammacell irradiator (Nordion). Following IR, cells were harvested at noted time points for IF or FISH on both settled cells and metaphase chromosome spreads.

## Generation of whole-chromosome oligopaints

Oligopaint libraries were designed as previously described, using the Oligoarray 2.1 software (*Beliveau et al., 2012*; *Beliveau et al., 2018*; *Rosin et al., 2018*) and the Dm3 genome build, and purchased from CustomArray. Whole-chromosome Oligopaints were designed to have 42 basepairs of homology, and a density of approximately one probe per kilobase. Coordinates for all Oligopaints can be found below.

Oligopaints were synthesized as previously described (*Moffitt and Zhuang, 2016*; *Rosin et al., 2018*). Briefly, probes were first PCR amplified using Taq DNA polymerase (Invitrogen). These PCR products were then in vitro transcribed using the HiScirbe RNA Synthesis kit (NEB), and converted to RNA:DNA duplexes by reverse transcription (Maxima H minus RT, Thermo) using unlabeled primers (IDT). RNA was removed by alkaline hydrolysis.

| Target | Chromosome | Start | End |
|---|---|---|---|
| X | X | 8603 | 22348002 |
| 2L | 2L | 5824 | 22767457 |
| 2R | 2R | 16362 | 20999406 |
| 3L | 3L | 21054 | 24399634 |
| 3R | 3R | 321 | 27799510 |

## Metaphase chromosome spreads preparation

To induce mitotic arrest, $2.5 \times 10^5$ cells were treated with 0.5 µg/ml demecolcine (Sigma-Aldrich) for 1 hr at 24 degrees. Cells were then pelleted by centrifugation for 5 min at 600 $g$ at room temperature and resuspended in hypotonic solution (250 ml of 0.5% sodium citrate), and incubated for 8 min. Following incubation, cells were placed in a cytofunnel and spun at 1,200 rpm for 5 min with high acceleration using a cytocentrifuge (Shandon Cytospin 4; Thermo Fisher Scientific). Spreads for FISH were immediately fixed in cold 3:1 methanol: acetic acid for 10 min, while spreads for IF were fixed with 4% PFA for 10 min. Following fixation, all slides were washed 3 times for 5 min in PBS-T (PBS with 0.1% Triton X-100).

## FISH with Oligopaints

For FISH on mitotic spreads: following fixation and PBS-T washes, slides were subjected to an ethanol row (3 min each in 70%, 90%, then 100% ethanol) at −20˚. Slides were then dried at RT for 48–72 hr. Following drying, slides were denatured in 2xSSCT/70% formamide at 72˚ for 2.5 min, and again subjected again to an ethanol row at −20˚. Subsequently, slides were dried for 10 min at room temperature before adding Oligopaints.

For FISH on settled interphase cells: slides were fixed in 4% PFA for 10 min at RT, followed by 3 × 5 min washes in PBS-T. Slides were then washes once in 2xSSCT for 5 min at RT, once in 2xSSCT/ 50% formamide at 92˚ for 2.5 min, and once in 2xSSCT/50% formamide at 60˚ for 20 min.

For all slides, primary Oligopaint probes in hybridization buffer (10% dextran sulfate/2xSSCT/50% formamide/4% polyvinylsulfonic acid (PVSA)) were then added to the slides, covered with a coverslip, and sealed with rubber cement. Slides were denatured on a heat block in a water bath set to 92˚ for 2.5 min, after which slides were transferred to a humidified chamber and incubated overnight at 37˚. 100 pmol of each probe was used per slide in a final volume of 25 µl.

Approximately 16–18 hr later, coverslips were removed with a razor blade, and slides were washed in 2 × SSCT at 60˚ for 15 min, 2 × SSCT at RT for 15 min, and 0.2 × SSC at RT for 5 min. Secondary probes (10 pmol/25 µl) containing fluorophores were then added to slides, again resuspended in hybridization buffer, and covered with a coverslip sealed with rubber cement. Slides were

incubated at 37° for 2 hr in a humidified chamber, followed by washes in $2 \times$ SSCT at 60° for 15 min, $2 \times$ SSCT at RT for 15 min, and $0.2 \times$ SSC at RT for 5 min. All slides were washed with Hoescht DNA stain (1:10,000 in PBS) for 5 min, followed by $2 \times 5$ min washes in PBS before mounting in Slowfade (Invitrogen).

## Immunofluorescence

For IF on both interphase cells and mitotic spreads, cells were fixed with 4% PFA for 10 min. Slides were then washed 3X in PBS-T for 5 min with gentle rocking. Subsequently, cells were permeabilized with 0.5% Triton X-100 in PBS for 20 min, then blocked in 5% non-fat milk in PBS-T (0.1%) for 1 hr at room temperature. 30 µl of blocking solution containing diluted primary antibodies was applied on the area of the slide containing fixed cells, covered with a coverslip, and incubated in a humidified chamber overnight at 4°C. The next day, slides were washed 3X for 5 min in PBS-T, with gentle rocking, followed by incubation with 30 µl of secondary antibodies diluted in blocking solution for 1 hr at room temperature in a dark humid chamber. Slides were again washed 3X for 5 min in PBS-T, with gentle rocking, and were then washed with Hoechst (1:10,000 in PBS) for 5 min to visualize nuclei. Finally, slides were washed 2X in PBS-T for 5 min before mounting in SlowFade (Invitrogen). IF on metaphase spreads was performed using the same protocol. Primary antibody dilutions were as follows: rabbit-anti PH3S10 (Millipore; 1:1000); mouse anti-alpha tubulin (Sigma; 1:50); chicken anti-CID (gift from Gary Karpen; 1:1000); rabbit anti-HOAP/Hip-Hop (gift from Yikang Rong; 1:200), rabbit anti-GFP (Invitrogen A6455, 1:200). Secondary antibody dilutions were as follows: 488 goat anti-mouse (Jackson Labs, 1:100); 488 goat anti-rabbit (Jackson Labs, 1:200); Cy3 goat anti-rabbit (Jackson Labs, 1:200), 647 goat anti-chicken (Fisher, 1:500).

## Imaging, quantification, and data analysis

Images of cultured cells were acquired at 24°C on a Leica DMi8 widefield fluorescence microscope, using a 1.4 NA 63x oil-immersion objective (Leica) and Andor iXon Ultra emCCD camera. The following filter cubes were used for image acquisition: DAPI, Y5, FITC, and RHOD. All images were processed and deconvolved using the Leica LAS-X 3.3 software with 3D Deconvolution, and exported as TIF files. Images were segmented and measured using a modified version of the TANGO 3D-segmentation plug-in for ImageJ as described above (*Ollion et al., 2013*). For interphase CT volume and contact measurements, nuclei were segmented using the 'Hysteresis' algorithm, and CTs were segmented using the 'Spot Detector 3D' algorithm. CT contact was defined as two CT objects with greater than 0.5 µm$^3$ colocalization. Statistical tests were performed using Prism seven software by GraphPad. Figures were assembled in Adobe Illustrator.

Before fixation, cells were counted using a Countess II FL Automated Cell Counter (Fisher), and viability was measured using Trypan Blue solution. All mitotic defects and translocations were manually quantified using deconvolved images. For rearrangements, each channel was analyzed alone and with all other channels to look for co-localization or color junctions of FISH probes. Fluorescent signal corresponding to color junctions that was 1) higher than background levels, 2) colocalized with DNA, and 3) present on both chromatids of a chromosome, was scored as a rearrangement.

## Calculating intra-chromosomal and inter-chromosomal interaction changes from Cap-H2 KD Hi-C data

We used Juicer (*Durand et al., 2016*) to obtain the observed matrix for each chromosome pair with Knight-Ruiz (KR) normalization at 5 kb resolution from both control and Cap-H2 knockdown Hi-C datasets obtained from *Li et al. (2015)*. For each chromosome pair matrix, an average KR-normalized signal was calculated by averaging all 5 kb bins comprising the matrix. Bins that had an NA value were excluded from this calculation. The intra-chromosomal and inter-chromosomal changes for each chromosome pair were calculated as (CapH2 average – WT average)/(WT average).

## Acknowledgements

We are extremely grateful to Mischa Li, Shane Harding, and Roger Greenberg for help and use of their X-ray irradiator. We also thank Kim McKim, Roger Greenberg, Elissa Lei, and members of the Joyce lab for helpful discussions and critical reading of the manuscript. Additionally, we thank Giovanni Bosco for providing us with the pMT-Cap-H2::GFP construct, Gary Karpen for providing the

CID antibody, and Yikang Rong for providing the anti-HOAP antibody. Some reagents were obtained from the Developmental Studies Hybridoma Bank and the Drosophila Genomics Resource Center. This work was supported by grants from the Pittsburgh foundation (KA2017-91787) and NIH (R35GM128903) to EJ.

## Additional information

### Funding

| Funder | Grant reference number | Author |
|---|---|---|
| Pittsburgh Foundation | KA2017-91787 | Eric F Joyce |
| National Institute of General Medical Sciences | R35GM128903 | Eric F Joyce |

The funders had no role in study design, data collection and interpretation, or the decision to submit the work for publication.

### Author contributions

Leah F Rosin, Conceptualization, Data curation, Formal analysis, Validation, Investigation, Visualization, Methodology, Writing—original draft, Writing—review and editing; Olivia Crocker, Data curation, Formal analysis, Visualization; Randi L Isenhart, Formal analysis, Investigation, Methodology, Writing—review and editing; Son C Nguyen, Resources, Supervision, Methodology, Project administration; Zhuxuan Xu, Resources, Methodology; Eric F Joyce, Conceptualization, Data curation, Formal analysis, Supervision, Funding acquisition, Validation, Investigation, Visualization, Methodology, Writing—original draft, Project administration, Writing—review and editing

### Author ORCIDs

Leah F Rosin (iD) https://orcid.org/0000-0002-2489-4016
Eric F Joyce (iD) https://orcid.org/0000-0002-0418-2804

### Decision letter and Author response

Decision letter https://doi.org/10.7554/eLife.49553.012
Author response https://doi.org/10.7554/eLife.49553.013

## Additional files

### Supplementary files

• Transparent reporting form DOI: https://doi.org/10.7554/eLife.49553.010

### Data availability

All data generated or analysed during this study are included in the manuscript and supporting files.

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
