## [Decision Letter]

Thank you for submitting your article "Chromosome territory formation attenuates the translocation potential of cells" for consideration by *eLife*. Your article has been reviewed by Kevin Struhl as the Senior Editor, a Reviewing Editor, and three reviewers. The following individuals involved in review of your submission have agreed to reveal their identity: Giovanni Bosco (Reviewer #1).

The reviewers have discussed the reviews with one another and the Reviewing Editor has drafted this decision to help you prepare a revised submission.

Summary:

As you will see, the reviewers were fairly divided in their assessment of this work. Two referees were very positive about the findings, and offered only minor suggestions to improve the presentation, while a dissenting opinion questioned the novelty of the work as well as some important aspects of the presentation. I have found it difficult to reconcile these disparate views, and have dithered in coming to a decision, for which I apologize. Ultimately, I feel that the story is quite interesting and establishes an important test of the significance of chromosome territories, but the work should be presented in a way that addresses the key caveats raised by reviewers. One key concern is that the only method currently available to modulate chromosome territories is to manipulate the levels of Condensin II; thus, it is not currently possible to demonstrate that the effects observed on translocation frequencies are directly due to altered CTs rather than other effects of condensin overexpression or depletion. I think this should be addressed more directly in the Discussion.

One reviewer also felt that the cytological measures of inter-CT contacts should be further supported by Hi-C analysis. However, this would require extensive additional work, and I think it would not markedly change the key conclusions. Moreover, the effects of modulating Condensin have previously been analyzed by Hi-C (albeit in different cells) in the cited work from the Corces lab (Li et al., 2015). I would thus encourage you to integrate their findings more directly into your presentation, perhaps by re-analyzing their data, and at the least by summarizing their findings.

Essential revisions:

As requested by reviewer #2, I think it is essential to avoid implying that depleting Condensin II levels leads to "loss" of territories, and to be more precise and quantitative in describing these effects throughout the manuscript. In addition, I ask that you address the criticisms raised in a response letter and in a revised manuscript to the extent that you are able.

Reviewer #1:

This study by Rosin and colleagues seeks to understand whether the 3D organization of chromosomes into spatially separated territories have any consequences on how broken chromosomes are repaired. Specifically, whether inter-chromosomal translocations are prevented by robust chromosome separation. This is an inherently interesting question, while it is also relevant to understanding mechanisms that give rise to genome instability and chromosomal rearrangements. Although previous studies have sought to show that physical proximity of specific genomic loci are prone to participate in recombination events leading to translocations, until now there has not been a way to modulate chromosome territories in living cells so as to ask directly how or if changes in chromosome spatial organization affects the occurrence of translocations. This study adds two innovative approaches that allow the authors to query unbiased whole genome recombination events with single cell resolution while also genetically manipulating chromosome territory formation/maintenance. First, they use Oligopaints to detect each of the major *Drosophila* chromosomes, and this allows them to detect simple and complex inter-chromosomal translocations on metaphase arrested cells. Second, by changing the levels of a key condensin II complex subunit, Cap-H2, the authors are able to modulate the degree to which cultures *Drosophila* cells form and maintain chromosome territories. Because Oligopaint probes allow single cell resolution and is amenable to quantitative analysis of 3D spatial separation, the authors determined how changes in spatial organization cause chromosomal rearrangements.

As expected, chromosomes with the largest amount of DNA suffered the most number of rearrangements. However, the authors also nicely demonstrate that the higher the level of chromosome territory contact between two different chromosomes the higher the likelihood that those two chromosomes participated in translocation events. Conversely, the chromosome pairs with the lowest territory interface contacts had the fewest translocation events. Interestingly, they report a linear relationship between frequency of chromosome territory contacts with incidence of translocations. This relationship was not true for chromosome territory volume overlap, suggesting that inter-chromosomal contacts at the periphery of territories are somehow more prone to damage and/or recombination. RNAi depletion of Cap-H2 resulted in more territory contacts and moderately more translocations, while increasing Cap-H2 levels led to highly separated chromosomes and a 35% decrease in the overall level of translocations. When specific chromosome pairs were analyzed separately, frequencies of translocations were decreased even further.

This is a very convincing, first of its kind study that demonstrates spatial separation of chromosomes indeed is biologically relevant and attenuates translocations after chromosome breakage. This study further implicates the condensin II complex and its interphase compaction activities as important factors that decrease the frequency of translocations by virtue of maintaining robust separation of chromosome territories. Given that condensins are conserved and essential in all species, this new insight into the interphase activities of condensin II and how they serve to protect the *Drosophila* genome from potentially deleterious chromosomal rearrangements should be of wide interest to many *eLife* readers. This reviewer is very enthusiastic about this study as it represents a highly significant advance in our mechanistic understanding of how 3D spatial organization of chromosomes directly impacts processes such as repair of broken chromosome and the origins of translocations. I found the manuscript to be concise, clearly written and without jargon. It was a pleasure to read.

Reviewer #2:

In this manuscript, Rosin and colleagues extend their work published in PLoS Genetics last year in which they show that in *Drosophila* loss of a condensin II component (CAPH2) alters the structure of chromosome territories in interphase cells.

They now go on to show in cell lines that this has some (modest) impact on the frequency at which chromosomal translocations are formed after irradiation. In its current form, I think that this manuscript submission is too preliminary and, as detailed below, the conclusions are over-stated.

Specific comments:

1) Throughout the manuscript the authors describe their condensin knockdown phenotype as "loss of chromosome territories". This is not correct – the CTs are not "lost". If the CTs were completely 'lost' the FISH hybridization signals would encompass the entire nucleus and there would be complete overlap between the three colors in Figure 3A. The authors should refer to their knockdown cells as having an "altered" CT conformation.

2) The authors interpret their data as evidence that it is the loss of CTs and the increased spatial overall between chromosomes in the condensin knockdown cells that is directly responsible for an elevated frequent of translocations. However, it is well known that chromatin structure at multiple levels affects the susceptibility to DNA damage and so it could also be that the altered chromatin structure in condensin knockdown cells makes the chromosomes more susceptibility to DNA damage, and that it is this that affects translocation frequency. The authors need to assess the level of endogenous DNA damage (γ-H2A.X staining) in control vs condensin knockdown cells.

3) The authors conclusion that there are increased interactions between chromosomes in condensin knockdown cells is based entirely on the interpretation of imaging data. The authors need to perform Hi-C on their control and condensin knockdown cells and to then quantify the increase in inter- vs intra-chromosomal contacts.

4) The authors need to be clearer about the limited spatial resolution of their imaging. Using wide-field epifluorescence the spatial resolution limit is going to be 200nm in x and y and 500nm in z. Therefore, any overlap between the hybridization signals has to be > than this before any claim can be made about overlaps between chromosome territories. For instance, I do not think that the data in Figure 2 can be used to conclude that (subsection “Inter-CT contacts correlate with translocation frequency in *Drosophila”*) 'CTs exhibit contact with each other in 93-99% of cells' and that a linear regression model for translocation frequency vs% CT contact can be reliably established with such low resolution imaging data. Quantitative CT contact data would come from Hi-C data (see point above).

5) Subsection “CT loss following knockdown of Cap-H2 leads to a moderate increase in translocation frequency”. It is overstating the data in Figure 3 to claim that knockdown of CapH2 leads to a 'moderate' increase in translocation frequency. For the most part (Figure 3F) there is no significant increase in translocation frequency scored between control and CAPH knockdown cells – it is only by adding all of the data together (no IR, 5 Gy and 20 Gy) that they can just squeak over the p<0.05 threshold for significance.

6) The authors need to do a better job of discussing the implications of their work to other systems. In the mouse a hypomorph of condensin II has been shown to have no effect on chromosome territories and chromatin compaction in the interphase nucleus. The genome instability in the mutant mice is shown to result from mitotic defects (Woodward et al., 2016). The same is true in humans (Martin et al., 2016).

7) Introduction. The authors refer to translocations that occur "naturally in the human population". However, the references cited – Arsuga et al., 2004; Holley et al., 2002 and Hlatky et al., 2002 – all refer to translocations induced by cell irradiation, i.e. they are not "naturally occurring". The correct citation for natural translocation frequencies in the human populations, and the relationship to chromosome organization is Bickmore and Teague, 2002.

Reviewer #3:

The manuscript titled "Chromosome territory formation attenuates the translocation potential of cells" by Rosin et al., examines the relationship between chromosome territory contacts and chromosomal translocations and addresses the question of whether the formation and maintenance of chromosome territories (CTs) is necessary to prevent chromosome translocations in cultured *Drosophila melanogaster* cells. The authors induce chromosomal translocations by irradiating diploid BG3 cells and then use Oligopaints to elegantly label the translocations. The authors present data showing that knockdown of the Condensin II subunit, Cap-H2, which had been previously shown to cause loss of CTs, results in small increases in translocation frequency for specific chromosome pairs. They also show that overexpression of Cap-H2 in these cells has the opposite effect and significantly inhibits translocations, even prior to irradiation. Combined, the data is convincing, thorough, and well-controlled, and they represent a significant advance in our understanding of the role that interphase chromosome territories play in the maintenance of genome stability.

---

## [Author Response]

Reviewer #1:This study by Rosin and colleagues seeks to understand whether the 3D organization of chromosomes into spatially separated territories have any consequences on how broken chromosomes are repaired. Specifically, whether inter-chromosomal translocations are prevented by robust chromosome separation. This is an inherently interesting question, while it is also relevant to understanding mechanisms that give rise to genome instability and chromosomal rearrangements. Although previous studies have sought to show that physical proximity of specific genomic loci are prone to participate in recombination events leading to translocations, until now there has not been a way to modulate chromosome territories in living cells so as to ask directly how or if changes in chromosome spatial organization affects the occurrence of translocations. This study adds two innovative approaches that allow the authors to query unbiased whole genome recombination events with single cell resolution while also genetically manipulating chromosome territory formation/maintenance. First, they use Oligopaints to detect each of the major Drosophila chromosomes, and this allows them to detect simple and complex inter-chromosomal translocations on metaphase arrested cells. Second, by changing the levels of a key condensin II complex subunit, Cap-H2, the authors are able to modulate the degree to which cultures Drosophila cells form and maintain chromosome territories. Because Oligopaint probes allow single cell resolution and is amenable to quantitative analysis of 3D spatial separation, the authors determined how changes in spatial organization cause chromosomal rearrangements.As expected, chromosomes with the largest amount of DNA suffered the most number of rearrangements. However, the authors also nicely demonstrate that the higher the level of chromosome territory contact between two different chromosomes the higher the likelihood that those two chromosomes participated in translocation events. Conversely, the chromosome pairs with the lowest territory interface contacts had the fewest translocation events. Interestingly, they report a linear relationship between frequency of chromosome territory contacts with incidence of translocations. This relationship was not true for chromosome territory volume overlap, suggesting that inter-chromosomal contacts at the periphery of territories are somehow more prone to damage and/or recombination. RNAi depletion of Cap-H2 resulted in more territory contacts and moderately more translocations, while increasing Cap-H2 levels led to highly separated chromosomes and a 35% decrease in the overall level of translocations. When specific chromosome pairs were analyzed separately, frequencies of translocations were decreased even further.This is a very convincing, first of its kind study that demonstrates spatial separation of chromosomes indeed is biologically relevant and attenuates translocations after chromosome breakage. This study further implicates the condensin II complex and its interphase compaction activities as important factors that decrease the frequency of translocations by virtue of maintaining robust separation of chromosome territories. Given that condensins are conserved and essential in all species, this new insight into the interphase activities of condensin II and how they serve to protect the Drosophila genome from potentially deleterious chromosomal rearrangements should be of wide interest to many eLife readers. This reviewer is very enthusiastic about this study as it represents a highly significant advance in our mechanistic understanding of how 3D spatial organization of chromosomes directly impacts processes such as repair of broken chromosome and the origins of translocations. I found the manuscript to be concise, clearly written and without jargon. It was a pleasure to read.

We thank this reviewer very much for their positive comments and are thrilled that they have enjoyed our story.

Reviewer #2:In this manuscript, Rosin and colleagues extend their work published in PLoS Genetics last year in which they show that in Drosophila loss of a condensin II component (CAPH2) alters the structure of chromosome territories in interphase cells.They now go on to show in cell lines that this has some (modest) impact on the frequency at which chromosomal translocations are formed after irradiation. In its current form, I think that this manuscript submission is too preliminary and, as detailed below, the conclusions are over-stated.Specific comments:1) Throughout the manuscript the authors describe their condensin knockdown phenotype as "loss of chromosome territories". This is not correct – the CTs are not "lost". If the CTs were completely 'lost' the FISH hybridization signals would encompass the entire nucleus and there would be complete overlap between the three colors in Figure 3A. The authors should refer to their knockdown cells as having an "altered" CT conformation.

We agree with the reviewer that this terminology is too vague to describe the phenotype following Cap-H2 KD. We have changed the wording throughout the manuscript to be more precise and, when possible, more quantitative. Specifically, we now use “increased CT intermixing” or “CT disruption” rather than CT loss. We do note however that, as shown in Figure 3C, the Oligopainted chromosomes do in fact decondense so severely as to essentially fill the entire euchromatic portion of the nucleus following Condensin II loss (the regions that remain unpainted are only the heterochromatin compartment and nucleolus, which the paints do not label; see Author response image 1).

2) The authors interpret their data as evidence that it is the loss of CTs and the increased spatial overall between chromosomes in the condensin knockdown cells that is directly responsible for an elevated frequent of translocations. However, it is well known that chromatin structure at multiple levels affects the susceptibility to DNA damage and so it could also be that the altered chromatin structure in condensin knockdown cells makes the chromosomes more susceptibility to DNA damage, and that it is this that affects translocation frequency. The authors need to assess the level of endogenous DNA damage (γ-H2A.X staining) in control vs condensin knockdown cells.

Thank you for bringing this accidental omission to our attention. We have now included the endogenous levels of DNA damage before IR in both control and Cap-H2 knockdown cells (added to Figure 3 —figure supplement 1). We do not see increased DNA damage after Cap-H2 RNAi alone.

3) The authors conclusion that there are increased interactions between chromosomes in condensin knockdown cells is based entirely on the interpretation of imaging data. The authors need to perform Hi-C on their control and condensin knockdown cells and to then quantify the increase in inter- vs intra-chromosomal contacts.

Thank you for the suggestion. As the editor as pointed out, this Hi-C data already exists for *Drosophila* Kc cells published by Li et al., 2015. Although the authors noted increased trans interactions, the quantification was not included in their manuscript. Therefore, we have now reanalyzed their Hi-C data for control and Cap-H2 KD cells, which shows a 7-17% increase in trans interactions between different chromosome arms (CTs). While it is difficult to make direct comparisons between population-averaged sequencing data and single-cell imaging data, we believe this fully supports our observation that inter-CT contacts are increased in the absence of Cap-H2. An accompanying graph along with the Hi-C subtraction map has now been included in Figure 3.

4) The authors need to be clearer about the limited spatial resolution of their imaging. Using wide-field epifluorescence the spatial resolution limit is going to be 200nm in x and y and 500nm in z. Therefore, any overlap between the hybridization signals has to be > than this before any claim can be made about overlaps between chromosome territories. For instance, I do not think that the data in Figure 2 can be used to conclude that (subsection “Inter-CT contacts correlate with translocation frequency in Drosophila”) 'CTs exhibit contact with each other in 93-99% of cells' and that a linear regression model for translocation frequency vs% CT contact can be reliably established with such low- resolution imaging data. Quantitative CT contact data would come from Hi-C data (see point above).

To address the spatial limit of resolution with deconvolved wide-field microscopy images, we used a threshold of signal colocalization greater than 500 nm^3^ (our 3D voxel volume) to define inter-CT contact. This threshold is described in the methods section, and we now have more specifically described how CT intermixing and inter-CT contact frequencies are calculated in the main text (subsection “Inter-CT contact frequency correlate with translocation frequency in *Drosophila*”).

Regarding Hi-C, there are numerous reports showing beautiful population-based correlations between relative translocation breakpoint usage and relative chromosome contact frequencies by Hi-C. Here, our main goal was to determine if absolute translocation frequencies are related to absolute frequencies of chromosome contacts and thus required single-cell information to address this question.

5) Subsection “CT loss following knockdown of Cap-H2 leads to a moderate increase in translocation frequency”. It is overstating the data in Figure 3 to claim that knockdown of CapH2 leads to a 'moderate' increase in translocation frequency. For the most part (Figure 3F) there is no significant increase in translocation frequency scored between control and CAPH knockdown cells – it is only by adding all of the data together (no IR, 5 Gy and 20 Gy) that they can just squeak over the p<0.05 threshold for significance.

We were initially struck by the reproducible increase in translocation frequency, albeit small, for each condition and replicate following Cap-H2 KD. However, we do note that none of these increases were significant on their own and, for clarity, we have removed the combined data from Figure 3. Instead, we highlight in Figure 3H that chromosome pairs 2 and 3 specifically do show a significant increase across all three replicates. Different sensitivities across different chromosome pairs was something we observed through this entire study, including for Cap-H2 overexpression, and most likely reflects the increased genomic length of the two chromosomes involved. Overall, the relative small increase in translocation frequency further supports our model that it is influenced more by inter-CT contact frequency, which is moderately changed, rather than the total extent of intermixing, which is dramatically increased following Cap-H2 KD. This idea and relationship to observations made in mammalian systems is discussed further in the Discussion section.

6) The authors need to do a better job of discussing the implications of their work to other systems. In the mouse a hypomorph of condensin II has been shown to have no effect on chromosome territories and chromatin compaction in the interphase nucleus. The genome instability in the mutant mice is shown to result from mitotic defects (Woodward et al., 2016). The same is true in humans (Martin et al., 2016).

We appreciate the reviewer pointing this out and agree that these mammalian phenotypes in NCAPH2 hypomorphs need to have been better discussed. We have included these references and summarized results in the Discussion section.

7) Introduction. The authors refer to translocations that occur "naturally in the human population". However, the references cited – Arsuga et al., 2004; Holley et al., 2002 and Hlatky et al., 2002 – all refer to translocations induced by cell irradiation, i.e. they are not "naturally occurring". The correct citation for natural translocation frequencies in the human populations, and the relationship to chromosome organization is Bickmore and Teague, 2002.

We thank this reviewer for bringing this to our attention and have now corrected the citation.